# Prevalence of *Toxoplasma gondii* in Endangered Wild Felines (*Felis silvestris* and *Lynx pardinus*) in Spain

**DOI:** 10.3390/ani13152488

**Published:** 2023-08-01

**Authors:** Pablo Matas Méndez, Isabel Fuentes Corripio, Ana Montoya Matute, Begoña Bailo Barroso, Rebeca Grande Gómez, Ariadna Apruzzese Rubio, Francisco Ponce Gordo, Marta Mateo Barrientos

**Affiliations:** 1Facultad de Veterinaria, Universidad Alfonso X el Sabio, Villanueva de la Cañada, 28691 Madrid, Spain; pmatamen@uax.es; 2Laboratorio de Referencia e Investigación en Parasitología, Centro Nacional de Microbiología, Instituto de Salud Carlos III, Majadahonda, 28220 Madrid, Spain; ifuentes@isciii.es (I.F.C.); begobb@isciii.es (B.B.B.); 3Departamento de Sanidad Animal, Facultad de Veterinaria, Universidad Complutense, 28040 Madrid, Spain; amontoya@ucm.es; 4Technical Assistence, General Direction of Natural Environment and Biodiversity, Ministry of Sustainable Development, Autonomous Community of Castilla-La Mancha, 45007 Toledo, Spain; rgrandegomez@gmail.com (R.G.G.); aarubio710@gmail.com (A.A.R.); 5Departamento de Microbiología y Parasitología, Facultad de Farmacia, Universidad Complutense, 28040 Madrid, Spain

**Keywords:** European wildcat, *Felis silvestris*, Iberian lynx, *Lynx pardinus*, *Toxoplasma gondii*, prevalence, epidemiology, immunofluorescence antibodies test, PCR, Spain

## Abstract

**Simple Summary:**

The wildcats (*Felis silvestris*) and Iberian lynx (*Lynx pardinus*) are species of special relevance in the Spanish fauna and are protected, as their populations are small and the lynx is classified as endangered. Both feline species can become infected by *Toxoplasma gondii*, a parasite that can cause morbidity and mortality in transplacentally infected or immunocompromised mammals. The data on the prevalence of this parasite in Spanish wild populations of wildcats and lynx were last reported 10 years ago, so the objective of this study is to update this information and assess the importance of these felines in the parasite’s epidemiology and the potential impact of the parasite on the conservation of these species. Wild populations show a high prevalence, necessitating the establishment of monitoring programs to assess the health status of these animals.

**Abstract:**

The wildcat (*Felis silvestris*) and the Iberian lynx (*Lynx pardinus*) are important species in Spain, considered as near-threatened and endangered, respectively. Both can be infected by *Toxoplasma gondii*, a parasite that can cause morbidity and mortality in transplacentally-infected or immunocompromised mammals. The data on the prevalence of this parasite in wild populations of these species in Spain are outdated. The objective of this study was to update information and evaluate the role of these felines in parasite epidemiology and the potential impact of the parasite on their conservation. Blood and fecal samples were collected from captured animals, as well as the tongue, diaphragm, and spleen, from animals killed in road accidents in central Spain. An indirect fluorescent antibody test (IFAT) was used to detect parasite antibodies in serum, microscopy and molecular analysis were used to detect oocysts in feces, and molecular analysis was used to determine the existence of tissue cysts. Seroprevalence was 85% in wildcats and 45% in lynx, and parasite DNA was detected in the feces of one wildcat and in tissue samples from 10 wildcats and 11 Iberian lynxes. These results highlight the epidemiological importance and high risk of *T. gondii* infection in animals and humans in the studied areas. Considering feline susceptibility to infection, monitoring programs are needed to assess the health status of wild felines.

## 1. Introduction

*Toxoplasma gondii* is a parasite that affects warm-blooded vertebrates from different habitats and regions, from the Arctic to the tropics, in terrestrial, aquatic, and marine environments [1]. This protozoan parasite causes toxoplasmosis, a disease of significant zoonotic importance in both human and animal contexts. *Toxoplasma gondii* infections in immunocompetent adult hosts are asymptomatic or subclinical; however, in immunocompromised or otherwise more susceptible individuals, it can cause severe disease with anorexia, weight loss, lethargy, dyspnea, ocular manifestations (chorioretinitis, blindness), neurological and systemic disorders, abortion, and even death [2]. The parasite forms tissue cysts in all hosts, but the sexual cycle is completed only in primary-infected felines. Oocysts are released in feline feces during a short shedding period [3,4], contaminating water, soil, and/or food. The life cycle of *T. gondii* is complex as a consequence of the diverse routes of transmission between individuals of the same and different species [5]. Inter-species transmission occurs by the oral route, in which herbivores become infected after ingestion mainly of oocysts, whereas infection in carnivores and omnivores can also occur through ingestion of tissue cysts in infected prey. Intra-species transmission is also possible during fetal development, lactation, or through sperm [6]. In addition to domestic cats, nearly 20 species of wild felines belonging to *Acinonyx* (cheetah), *Felis* (“cats”), *Leopardus* (ocelot, Geoffroy’s cat, Pampa’s cat), *Herpailurus* (jaguarundi), *Lynx* (lynxes), *Panthera* (lion, tiger), and *Puma* (cougar) have been reported to shed oocysts of *T. gondii* [7,8]. As definitive hosts, felines (both domestic and wild) play a crucial role in the epidemiology of toxoplasmosis in all species and can be responsible (at least, domestic cats) for outbreaks [5,9], and it is essential to determine the prevalence of infection in these animals to implement appropriate control measures.

The majority of epidemiological studies in felines are based on determining seroprevalence in the analyzed population by detecting specific anti-*Toxoplasma* IgG antibodies [10,11,12]. This allows for the identification of infected individuals but does not determine whether they are in the acute phase (intestinal, with oocyst-release in feces) or in the chronic phase (with tissue cysts). In Spain, several studies have been carried out in domestic felines (stray and owned cats), revealing seroprevalence rates ranging from 12% to 84.7% [13,14,15,16,17,18,19,20]; however, oocysts have not been detected in any of the fecal analyses performed. Epidemiological information is more limited for the two native species of wild felines in Spain, namely the wildcat (*Felis silvestris*) and the Iberian lynx (*Lynx pardinus*) [21,22,23,24].

The wildcat has a wide distribution in Europe, and although its populations are increasing in certain Central European regions [25], the Iberian populations are declining [26] and are currently limited to a restricted geographical area [27]. The Iberian lynx is only found in the Iberian Peninsula and is an emblematic species of high ecological value. Although its population experienced a significant decline in the past, with only 93 individuals recorded in 2002 [28], thanks to various conservation projects, 1668 individuals were counted in 2022 [29]. Both feline species are listed as specially protected wild species under Spanish legislation (RD 139/2011 for the development of the List of Wild Species under Special Protection and the Spanish Catalog of Endangered Species), with the Iberian lynx classified as endangered. There are many potential threats to these species, including habitat destruction and fragmentation, decline of prey, hybridization, hunting, roadkill, and infectious and parasitic diseases [30,31,32,33,34]. Within parasitic infections, those caused by *Toxoplasma gondii* in wild felines seem to be rarely associated with disease, and no cases of clinical toxoplasmosis have been described in Iberian lynx or wildcats. However, clinical cases have been reported in Pallas’s cats (*Otocolobus* [*Felis*] *manul*), and neonatal toxoplasmosis has been documented in sand cats (*Felis margarita*) and bobcats (*Lynx rufus*) [35,36,37].

The objective of the present study was to evaluate the presence and prevalence of *T. gondii* in wild populations of Iberian lynx and wildcat in Spain in order to assess the epidemiological importance of these hosts and evaluate the potential impact of the parasite on the conservation of these protected species.

## 2. Materials and Methods

### 2.1. Study Area and Animals

Over a period of 24 months (January 2021 to December 2022), a total of 69 samples from Iberian lynxes and 20 samples from wildcats were collected in 7 provinces of two autonomous communities (AC) in the central region of Spain (Figure 1). The majority of the wildcat samples were obtained from the AC of Castilla y León, located in the central–northern region with a climate characterized by temperate summers and the absence of a dry season in some provinces. The lynx samples were collected from the AC of Castilla-La Mancha, situated in the central–southern part of the country with a more extreme climate characterized by dry summers and high temperatures. As part of the LIFE 19NAT/ES001055 LINXCONNECT project (https://lifelynxconnect.eu/en/project/; accessed on 28 July 2023), nine of the analyzed lynxes were captured animals undergoing health evaluations according to the protocols approved in 2012 within the LIFE 10NAT/ES/570 IBERLINCE project (http://www.iberlince.eu/index.php/eng/; accessed on 28 July 2023) (protocols available in Spanish, Portuguese, and English at http://www.iberlince.eu/images/docs/3_InformesLIFE/ProtocolosSanitarios_M.Iberlince.pdf; accessed on 28 July 2023). These animals were handled by authorized veterinarians following the established protocols within the mentioned conservation program. The remaining sampled animals (60 lynxes and 20 wildcats) were individuals found dead due to road accidents and were collected and frozen within 24 h of the estimated time of death by wildlife recovery centers under the authority of the Regional Environmental Departments of the respective AC. The handling of the animals and collection of samples were conducted by authorized veterinarians from Regional Environmental Departments. The samples were provided to us in accordance with the conditions stipulated in the permits issued by each Regional Environmental Department (references DGPFEN/SEN/avp_21_103_bis in Castilla-La Mancha, AB/is. Exp.AUES/CYL/001/2021 in Castilla y León). The age of the felines was estimated based on size, weight, and dental wear and classified into juveniles (<1 year), adults (1–7 years), and seniors (>7 years) [28,38].

### 2.2. Sample Collection and Processing

Blood samples were collected from live animals through venipuncture and from deceased animals from the heart; in all cases, thoracic fluid was also collected in case it would be needed. Samples were centrifuged at 1500 rpm for 10 min, and the sera were stored at −20 °C until analysis. Fecal samples were collected from the rectum of live animals and from the final section of the large intestine in necropsied animals. In both cases, they were kept refrigerated (4 °C) until analysis. Additionally, samples of the tongue, diaphragm, and spleen were collected from the deceased animals. These samples were stored at −20 °C until analysis using molecular techniques.

To detect *T. gondii*-specific antibodies, the indirect immunofluorescence antibody test (IFAT) was conducted according to the method described by [40] using a commercial IFAT kit for anti-*T. gondii* IgG detection (MegaFLUO^®^ *Toxoplasma g*., Megacor Diagnostik GmbH, Hörbranz, Austria) following the manufacturer’s instructions. The same kit was used in combination with anti-feline IgM conjugate (Fuller Laboratories, Fullerton, CA, USA) to determine IgM levels. The sera were screened at dilutions of 1/40, 1/80, 1/160, 1/320, and 1/640. An IgG or IgM titre of ≥1/80 was considered positive.

The fecal samples were processed using the modified Telemann method [41,42] for optical microscopy observation; molecular analysis was also performed. Using the Telemann method, approximately 3 g of feces were diluted in 20 mL of 5% acetic acid and filtered through gauze; the filtrate was mixed with 20 mL of ether and centrifuged at 1500 rpm for 5 min. One drop of the final sediment was observed in unstained preparations at magnifications of 400–1000×. For molecular analysis, DNA was extracted using the QIAamp^®^ Fast DNA Stool Mini Kit (Qiagen, Hilden, Germany) following the manufacturer’s instructions. Parasite detection was performed using two methods: (1) a nested PCR (nPCR) was conducted to amplify a 97 bp fragment from the B1 gene of *T. gondii*, following the protocol described by [43] using 5 and 15 µL of DNA from each sample; (2) a real-time PCR (qPCR) was performed to amplify the non-coding repetitive DNA fragment of 529 bp (RE 529-bp) of *T. gondii*, following the protocol described by [44] using 5 µL of DNA from each sample. A positive result in either of the two techniques was considered positive.

The tissue samples were first subjected to enzymatic digestion to release bradyzoites from the tissue cysts in order to obtain more parasite DNA and reduce the risk of false negatives. Following the protocol described by [45], digestion was performed using Trypsin-EDTA 1X in PBS (Biowest, Nuaillé, France), using 2 g of each sample. Subsequently, DNA extraction was carried out using a robot (QIAcube) and the QIAamp^®^ DNA Mini Kit (QIAGEN, Hilden, Germany), following the manufacturer’s instructions. The extracted DNA was then used for nPCR and qPCR under the same conditions as indicated previously for fecal sample analysis.

### 2.3. Statistical Analyses

Seroprevalence was defined as the percentage of samples testing positive for antibodies to *T. gondii*, with 95% confidence intervals established. The Chi-square test was employed to determine significant associations between animal species (*F. silvestris* vs. *L. pardinus*). Comparisons were conducted for age groups (juvenile, adult, and senior), sex (male vs. female), and geographic area to assess differences in *T. gondii* seroprevalence among each species and as a combined group (wild felines). Differences were considered statistically significant at *p* ≤ 0.05. The concordance between serodiagnostic, coprological, and molecular analysis results was assessed using the kappa index. The statistical tests were performed using SPSS Statistics ver. 25 (IBM, New York, NY, USA).

## 3. Results

Of the 89 felines sampled, the majority were adult animals (84% and 100% in Iberian lynx and wildcats, respectively), and there were no wildcat juveniles. The proportion of males was higher (67 and 80%) than that of females (33 and 20%) in Iberian lynx and wildcats, respectively.

In the analyzed fecal samples of wildcats and Iberian lynx, oocysts compatible with *T. gondii* were found in one wildcat.

The analysis of the sera detected IgG antibodies against *T. gondii* in 53.9% (95% CI: 43.5–64.2) of the analyzed wild felines, with a significantly higher seroprevalence (*p* = 0.002) observed in wildcats (85%, 17/20; 95% CI: 69.3–100) compared to Iberian lynx (44.9%, 31/69; 95% CI: 33.1–56.6) (Table 1). IgM antibodies were detected in one wildcat, which also tested positive for IgG (IgM titre 1/160, IgG titre 1/640).

In the molecular analyses, parasite DNA was detected in the feces of one wildcat (the individual seropositive for IgM and also IgG and positive by microscopy) and in tissue samples from 10 wildcats and 11 Iberian lynxes (Table 2); no positive samples were found in seronegative animals. The detection of *T. gondii* in tissues was higher when considering the combined results of nPCR and qPCR, as there were many samples that tested positive with one technique and negative with the other; four samples from wildcats and one sample from Iberian lynx tested positive with both PCR methods. The highest overall number of PCR-positive samples was found in the tongue (11), followed by the spleen (9), and the diaphragm (4), with nPCR being the more sensitive molecular method (17 positives: 10 in the tongue, 6 in the spleen, and 3 in the diaphragm) compared to qPCR (9 positives: 5, 3, and 2, respectively). Most animals in which *T. gondii* was detected by PCR had an antibody titre equal to or greater than 1/640.

Regarding the analyzed demographic variables associated with *T. gondii* infection, statistically significant differences were found regarding age, with higher seroprevalence in the adult (56.4%, 44/78; 95% CI: 45.4–67.4) and senior groups (100%; 4/4), compared to the young group (0%; 0/7); however, no statistically significant difference was found in relation to sex (Table 3). The provinces with the highest number of samples were Burgos in Castilla y León (wildcats) and Toledo in Castilla-La Mancha (Iberian lynx) (Table 3).

## 4. Discussion

This is the first study to report the prevalence of *T. gondii* in wildcats in Castilla y León, analyzing a larger number of animals compared to previous studies conducted in other regions of Spain [21,34]. Additionally, analyzed a greater number of lynxes in Castilla-La Mancha than in previous studies (Table 4). Furthermore, the availability of animals for epidemiological studies in these species is typically low and limited to specific territories. Most epidemiological studies on *T. gondii* in Iberian lynx focused on populations in Andalusia [22,23,24], with only one study conducted in Castilla-La Mancha [21]. Similarly, studies on the wildcat have been limited to a small number of animals (six to nine individuals) from the Cantabrian Mountains [21] and Castilla-La Mancha [34]. Our study provides updated data on *T. gondii* infection in the Iberian lynx, considering that more than 10 years have passed since the last published studies, and the estimated population size has changed, increasing from 213–241 individuals (2008–2009) to over 1500 individuals currently [29]. This increase in population size results in greater interaction among individuals and different territorial distribution.

Given the short patent period of *T. gondii* infections in felines (1–2 weeks following primary infection) [46], it is rare to find oocysts in fecal samples from these hosts. The low prevalence of positive fecal samples for oocysts found in this study (5%, 1/20, in wildcats) is similar to other studies on wild felines, including red lynx (*Lynx rufus*) (6.2%; 1/16) and cougars (*Puma concolor*) (1.9%; 1/52) in North America [8], jaguars (*Puma yagouaroundi*) (1.22%; 1/82) in a regional park in Brazil [47], and wildcats (*F. silvestris*) from Greece (1.6–4.3%) [48]. Oocysts can be difficult to differentiate from those of *Hammondia hammondi* and *Besnoitia* spp. [49,50,51]; therefore, identification should be confirmed through molecular analysis. In the present study, the positive fecal sample in a wildcat was confirmed through molecular analysis. All diagnostic results from this individual are suggestive of active infection with oocyst shedding. This is the first report in Spain of *T. gondii* oocysts in a wild feline (wildcat) in its natural environment. In the case of the Iberian lynx, the negative results for the presence of oocysts in feces or from PCR analysis of fecal samples prevent confirming this species as a definitive host of *T. gondii*. This negative result is not surprising; due to the short patent period, it is difficult to detect oocysts, even in cat feces [52]. However, the diversity of felid species accepted as definitive hosts, including congeneric species (*L. lynx*, *L. rufus*) [7,8] on that list, strongly suggests that the Iberian lynx may also be a definitive host. Further studies are needed to undoubtedly confirm this possibility.

There are few studies that assess the prevalence of *T. gondii* in wild felines worldwide, and most of them employ serology as a diagnostic method. Several methods have been used for the serological diagnosis of *T. gondii* in wild animals (ELISA, MAT, IFAT, IHA, latex agglutination test, and immunochromatographic tests) [53,54]. The IFAT method used in the present study has the same sensitivity and specificity of ELISA and is better than MAT [55]. The seroprevalence of *T. gondii* detected in wild felines is 59% globally and 67% in Europe [53]. These results are similar, although slightly higher, to those obtained in our study, where IgG antibodies against *T. gondii* were detected in 53.9% of the samples. However, comparing the results of different studies can be challenging in many cases due to differences in the techniques employed, including variations in sensitivity and specificity, sample size, variability in terms of the age, and origin of the analyzed felines, as well as geographical and habitat differences, even within the same country.

The seroprevalence detected in the samples of lynxes in our study (44.9%) is similar to that detected using the IHA technique in lynxes from Andalusia (44%) [22] but lower than that obtained in studies where the MAT (Modified Agglutination Test) technique was used (63–82%) [21,23,24]. The seroprevalence results obtained in the wildcats in our study are very high (85%) compared to the data published in other studies conducted in Spain (50–56%) [21,34]. The differences may be related to the small number of animals analyzed in these studies (Table 4) and the different geographical areas studied. The prevalence of this parasite in humid areas may be higher than in dry or desert regions due to the viability of oocysts in these environmental conditions [1,56]. We found a higher prevalence in animals inhabiting the northern regions, where a subhumid continental climate with milder temperatures and no dry season prevails, compared to the southern regions, where a warm continental climate with extreme temperatures and dry summers prevails. However, besides climate, there are other factors that need to be considered to explain the differences in prevalence, especially when comparing different host species. In the case of *T. gondii*, the definitive host can become infected through the ingestion of oocysts (whose viability depends on environmental factors). However, although possible, the ingestion of such a high number of oocysts would be required, making this route of infection highly inefficient [57]; it is considered that the parasite is primarily transmitted to the definitive host through predation [6,57]. In this case, it is necessary to take into account the differences in dietary preferences between wildcats and lynxes. The European rabbit (*Oryctolagus cuniculus*) is the preferred prey of the Iberian lynx [58]; although, in the absence of these animals, they may prey on other species (e.g., hares, *Lepus* sp.). In contrast, the diet of wildcats is not as specific and depends more on the availability of prey (rabbits or rodents) [59]. Additionally, the size of the prey is important for two reasons. Firstly, it affects the number of individuals that need to be captured for the predator to obtain an adequate amount of food, and more prey means a higher risk of infection. In the necropsy findings of this study, an average of 3–7 mice were found in the stomachs of wildcats, while in the stomachs of lynxes, which are strictly dependent on European rabbits, remains of 1–2 rabbits were found (P. Matas Méndez, personal observation). The wildcat, by preying on a greater number of intermediate hosts and consuming hundreds of rodents throughout its life, increases the probability of becoming infected. In fact, there are studies that have observed a correlation between prevalence rates in felines and the number of rodents consumed, which varies according to prey abundance, time of year, and local conditions [13]. Secondly, some authors have suggested that since the prevalence of infection increases with age, larger prey (rabbits, hares) with a longer lifespan than smaller prey (rodents) are more likely to be infected [60]. In addition to lifespan, the risk of infection also depends on the susceptibility of each species [61,62]. We have not found published studies on the seroprevalence of *T. gondii* in rodents in Spain; in other European countries, it varied in the range of 0–59.4% [63]. In other host species from Spain, seroprevalence varied in the range of 5.6–39.6% in ruminants (roe deer, fallow deer, red deer, Sothern chamois, mouflon, wild goat, and Barbary sheep) [64], 21.9% in free-ranging Iberian pigs [65], and 6.1–53.8% in wild rabbits [66].

Although the prevalences found in wildcats and Iberian lynxes were slightly higher in males than in females, the difference is not statistically significant. These results are consistent with other studies where this variable is also not considered significant, as both sexes have the same risk of exposure to infection sources [53]. However, some studies conducted in feral cats observed greater predation by males and, therefore, a higher risk for infection [60]. Only adult wildcats have been analyzed in this study, but lynxes from all three age groups were analyzed, although there was a predominance of adult individuals (58) compared to juveniles (7) and seniors (4). In lynxes, we observed a significant difference among age groups, with an increase in individuals over 1year old (adults and seniors). This result is common in all studies that consider this variable, as the definitive host also has a higher probability of having been infected with increasing age [1,60]. Regarding the geographical distribution, the significant disparity in the number of wildcats collected across different provinces, with only 1–3 animals sampled in most of them, precludes drawing conclusions regarding this variable in this host species. Furthermore, there is no reliable population census available, as wildcat is not considered an endangered species but rather classified as near-threatened, which diminishes official interest and the publication of studies. In the case of the Iberian lynx, the samples primarily originate from two areas, one in the province of Toledo and another in Ciudad Real, corresponding to the distribution of the populations reported in the latest census [29]. In the province of Albacete, only two lynxes were analyzed; however, considering that the estimated population in this province was six individuals, it represents a significant percentage of animals. Overall, the observed seroprevalence for lynx is similar across the different studied areas, averaging around 45%. This value is at the lower limit of the range reported in other studies in Spain (Table 4). To the best of our knowledge, there are no available published data from Portugal.

In this study, *T. gondii* infection in wild felines was confirmed by detecting parasite DNA in different tissues (tongue, diaphragm, and spleen) analyzed using nPCR and qPCR. Parasite DNA was detected in only 26.2% (21/80) of the analyzed felines (tissue samples were not obtained from nine felines as they were live animals captured for monitoring). Despite 53.9% (48/89) of the wild felines showing antibodies (IgG) against *T. gondii* (indicating past infection and the presence of cysts in their tissues), parasite DNA was only detected in 43.7% (21/48) of the seropositive felines. Other studies also reported higher prevalences by serology than by detection of parasite DNA [67,68]. It is challenging to determine the optimal tissue sample for PCR diagnosis, as cysts can be found in different tissues, with a particular tropism for the central nervous system and other highly oxygenated tissues. Some studies compared DNA detection in different tissues and indicated the heart and brain as the preferred tissues [55,69,70,71]. However, in other studies, higher detection was observed in tongue samples [72], brachiocephalic muscle, and even in the spleen, with the same frequency as the brain [73]. In this study, access to samples of certain tissues from the roadkill animals was not possible due to the condition in which some of the remains were found. Among the tissues/organs that could not be analyzed in all animals was the brain. The selection of the tongue and spleen was based on the literature, and the choice of the diaphragm was based on the well-oxygenated nature of this muscle. We detected a higher number of positives in the tongue, followed by the spleen and diaphragm. However, we believe that these results should be compared with findings from other tissues and that the sample size should be increased to assess the tropism and sensitivity in detecting *T. gondii* in the tissues of wild felines. The PCR positivity depends on the number of cysts in a single animal and in a single organ/tissue; in food animals, it is estimated that as few as one tissue cyst may be present in 100 g of meat [74]. In the present study, only 2 gr of each organ/tissue were analyzed; as a result, false negatives can occur due to a non-homogeneous distribution of *T. gondii* cysts in the tissues and/or a low number of cysts, so when using a small sample for DNA detection, there may not have been cysts in the processed fragment (and therefore no detectable parasitic DNA). Some authors propose to address this issue using the magnetic capture PCR (mcPCR) method, as it allows for the analysis of a larger amount of tissue than conventional methods, concentrating *T. gondii* DNA and increasing sensitivity [75,76,77].

## 5. Conclusions

This is the first study in which *T. gondii* oocysts were found in fecal samples from the wildcat, *Felis silvestris*, in Spain. Oocysts were not detected in the feces of Iberian lynxes, *Lynx pardinus*; this was an expected result even in cats, so it is not possible to confirm or deny at this time whether lynxes are definitive hosts of *T. gondii.* The *T. gondii* seroprevalence found in wildcats in this study is 85%, and in Iberian lynx, it is 45%; in this species, an increase in seroprevalence with the age of the individuals was observed. The higher seroprevalence in wildcats than in Iberian lynxes may be due to the different geographic locations of the two feline populations, as well as differences in their diet. The detection of the parasite in feces or in tissue samples was low compared to the high seroprevalence detected in the same individuals. The high number of infected animals alerts us to the potential degree of environmental contamination and the risk to other animals and humans. It is necessary to establish surveillance and control programs for these wild felines, as well as for the prey species that are part of their food chain, to protect the health of these feline populations, other animals, and public health. Studies comparing the sensitivity of different diagnostic techniques, as well as the tissues to be analyzed, are necessary to identify the tissues with a higher tropism in these host species, as it may be beneficial for future epidemiological studies.

## Figures and Tables

**Figure 1 animals-13-02488-f001:**
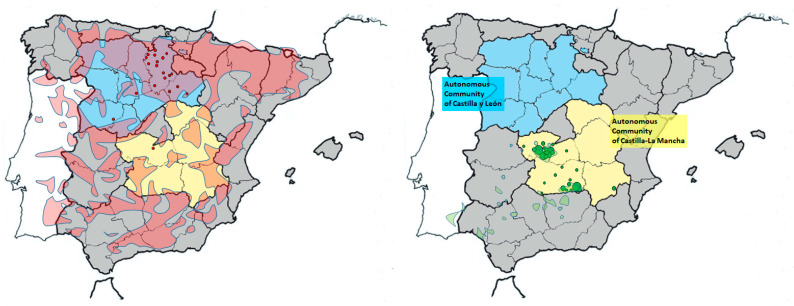
Maps of the Iberian Peninsula showing the two Spanish autonomous communities where the samplings were conducted. The approximate location of origin of samples corresponding to wild cats (red circles, left map) and Iberian lynx (green circles, right map) obtained in each province are indicated. Circle sizes are proportional to the number of samples from that location. Distribution areas of wildcats (pale red) [39] and Iberian lynx (pale green) [29] are also shown.

**Table 1 animals-13-02488-t001:** Seroprevalence of *Toxoplasma gondii* in wildcat (*Felis silvestris*) and Iberian lynx (*Lynx pardinus*) obtained by indirect immunofluorescence antibody test (IFAT).

IFAT IgG	*Felis silvestris* *n* (%; 95 CI ^1^)	*Lynx pardinus* *n* (%; 95% CI)	*p*-Value
Negative	3 (15.0; 0.0–30.6)	38 (55.1; 26.5–49.4)	0.001
**Positive—any titre**	**17 (85.0; 69.3–100.0)**	**31 (44.9; 33.1–56.6)**	**0.002**
Positive—titre 1/80	---	4 (5.8; 0.4–11.6)	0.006
Positive—titre 1/160	1 (5.0; 0.0–14.5)	5 (7.2; 0.9–13.0)	0.724
Positive—titre 1/320	4 (20.0; 0.0–12.6)	6 (8.7; 2.2–15.7)	0.158
Positive—titre ≥ 1/640	12 (60.0; 38.5–81.5)	16 (23.2; 13.1–32.9)	0.001

^1^ Confidence interval.

**Table 2 animals-13-02488-t002:** Comparison of the results for *Toxoplasma gondii* identification in samples from wildcat (*Felis silvestris*) and Iberian lynx (*Lynx pardinus*) using serology (indirect immunofluorescence antibody test, IFAT), coprology (Telemann, nPCR, and qPCR), and tissue analysis (nPCR and qPCR).

Host Species	Serology (*n* = 89)		Coprology (*n* = 89)			Tissue Analysis (*n* = 80)		
	IFAT IgG	IFAT IgM	Telemann	nPCR	qPCR	nPCR ^1^	qPCR ^1^	Total PCR+ (Any Method) ^2^
*Felis silvestris*	17 ^3^	1 ^3^	1 ^3^	1 ^3^	1 ^3^	8 ^3^	6	10
*Lynx pardinus*	31	0	0	0	0	9	3	11
Total (both species)	48	1	1	1	1	17	9	21

^1^ These columns indicate the number of animals found positive with each method. Some animals were found positive by both methods; due to this, the sum of the values of these columns is greater than the values provided in the last column (total number of animals found positive by PCR, any method). ^2^ The values in this column indicate the number of animals found positive in at least one of the PCR tests (qPCR and/or nPCR in fecal and/or tissue samples). ^3^ One animal tested positive in IFAT (both antibodies), microscopy, and molecular analysis (feces and tongue samples).

**Table 3 animals-13-02488-t003:** Seroprevalence (positive by IFAT) in wild populations of wildcat (*Felis silvestris*) and Iberian lynx (*Lynx pardinus*) in relation to animal sex, age, and sample collection area.

	*Felis silvestris*			*Lynx pardinus*			Combined Data from Both Species		
Variable	*n*+	% Positive	*p*-Value	*n*+	% Positive	*p*-Value	*n*+	% Positive	*p*-Value
Sex			0.347			0.864			0.795
Male	13	81.2 (13/16)		21	45.6 (21/46)		34	54.8 (34/62)	
Female	4	100 (4/4)		10	43.5 (10/23)		14	51.9 (14/27)	
Age			n/a			0.002			0.003
Juvenile	---	---		0	0.0(0/7)		0	0.0 (0/7)	
Adult	17	85.0 (17/20)		27	46.5 (27/58)		44	56.4 (44/78)	
Senior	---	---		4	100.0 (4/4)		4	100.0 (4/4)	
Capture area									0.015
Castilla-La Mancha			n/a			0.919			
Albacete	---	---		1	50.0 (1/2)		1	50.0 (1/2)	
Ciudad Real	0	---		10	41.6 (10/24)		10	41.7 (10/24)	
Toledo	0	0.0 (0/1)		20	46.5 (20/43)		21	45.5 (21/44)	
Total	0	0.0 (0/1)		31	44.9 (31/69)		31	44.3 (31/70)	
Castilla y León			<0.001			---			
Burgos	13	92.9 (13/14)		0	---		13	92.9 (13/14)	
Salamanca	0	0.0 (0/1)		0	---		0	0.0 (0/1)	
Soria	3	100.0 (3/3)		0	---		3	100 (3/3)	
Valladolid	1	100.0 (1/1)		0	---		1	100 (1/1)	
Total	17	89.5 (17/19)		0			17	89.5 (17/19)	

**Table 4 animals-13-02488-t004:** Studies on the seroprevalence of *Toxoplasma gondii* in wild felines (*Felix silvestris* and *Lynx pardinus*) in Spain.

Host Species	Region	Animals Examined	Serological Method	% Positive	Reference
*Felis silvestris*	Andalusia, Castilla-La Mancha, Cantabrian region	6	MAT (modified agglutination text)	50	[21]
	Castilla-La Mancha	9	ELISA	56	[34]
	Castilla-La Mancha, Castilla y León	20	IFAT	85	Present study
*Lynx pardinus*	Andalusia, Castilla-La Mancha	27	MAT	82	[21]
	Andalusia	48	IHA (indirect haemagglutination)	44	[22]
	Andalusia	26	MAT	81	[23]
	Andalusia	129	MAT	63	[24]
	Castilla-La Mancha	69	IFAT	45	Present study

## Data Availability

All new data are presented in this study; data sharing is not applicable to this article.

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
