# Peer review of "Prevalence of *Toxoplasma gondii* in Endangered Wild Felines (*Felis silvestris* and *Lynx pardinus*) in Spain"

_animals, 2023, doi:10.3390/ani13152488_

Round 1
Reviewer 1 Report
Overall, the manuscript is well written - minor revisions needed only. The research was well-designed. As this is a zoonotic parasite and your results suggest a high risk of exposure in the areas studied, perhaps a statement in the discussion/conclusion is warranted to emphasize the public health implications.
See attached for detailed feedback

Author Response
General feedback
Overall, the manuscript is well written. The research was well-designed. In instances where "genetic" testing is stated, it may be more appropriate to state "molecular" testing (in reference to the PCR tests performed). There were instances where the number of positive on serology is compared to that using molecular techniques - caution must be used in "compari ng" these diagnostic tools as serology (lgG) confirms previous infection/exposure. In the case of Toxoplasma, lgG may/may not suggest establishment of tissue cysts. PCR, on the other hand, is used to detect the parasite (active infection). lt would be expected that seroprevalence is higher than PCR - which makes direct comparison challenging. The use of lgM diagnostics in this study was a good idea as this form of serology is more suggestive of an active infection (vs. lgG).
Thanks very much for the time you have dedicated to this review, for your comments and for the help with the use of the English language.
We have changed genetic -> molecular as suggested. In relation to the comparisons between the results of serological and molecular biology analyses, it is a very good comment. The reviewer is correct regarding the basic concept of what the results of each method imply. In theory, there could be cases of individuals in which T. gondii has not developed into tissue cysts because immunity has eliminated it before, but antibodies have formed due to activation of adaptive immunity. In this case, prevalence (results from molecular analyses) and seroprevalence would be very different concepts, as prevalence would indicate the number of true parasitized individuals in the population, while seroprevalence would indicate the number of individuals who have been infected at some moment in the past but may not be currently parasitized. However, the infectivity of bradyzoites in cats is very high, with as few as 50 bradyzoites (range 2-181) being sufficient to produce tissue cysts and intestinal infection (Cornelissen et al., 2014, PLOS One 9: e104740) and 100 oocysts to produce tissue cysts (Dubey 1997, Parasitology 115: 15-20). Considering that a tissue cyst harbors between 2-1000 bradyzoites (Dubey et al., 1998, Clin. Microbiol. Rev. 11:267-299), it can be assumed that a cat (and by extension, other felids) feeding on infected intermediate hosts (the normal route of transmission) or ingesting an enough number of oocysts (100, maybe less) will develop tissue cysts.
A second aspect to consider is the survival of cysts within the host’s tissues. It has been considered that the infection persists for the lifetime of the host (Butler et al. 2013, Trends Parasitol. 29: 593-602). Although this is still to be definitively demonstrated (Rougier et al. 2016, Trends Parasitol. 33: 93-101), there is no data to dismiss the widespread belief of long-term survival of tissue cysts. Other different question, outside the scope of our work, is if the chronic infection generates a protective immune response against reinfection. Rougier et al. (2016) did not questioned the presumed life-lasting survival of the tissue cysts but the dogma of “once infected, forever protected” as there have been cases of congenital infection in the offspring of immunocompetent mothers with positive serology before pregnancy.
For the purposes of the present study, we think it can be accepted that seropositive animals have tissue cysts, and prevalence (results from molecular analyses) and seroprevalence can be compared.
Simple summary
Line 26: suggest removing the first "their" and replace with "the" and remove the 2•• "their"
The text has been changed and these words have been removed; the new text (line 26) is "a parasite that can cause morbidity and mortality in transplacentally infected or of immunocompromised mammals."
Line 27: suggest replacing "•.• are a decade old•.•" with "...were last reported 10 years ago•.•"
Done as requested (line 28): "...were last reported 10 years ago, ..."
Line 31-33: see comment in General feedback regarding discussion of serology vs molecular testing. Would suggest removing statement.
As commented above, we think that results from serology and PCR can be compared. Anyway, this paragraph has been removed as suggested.
Abstract
Line 35: lnconsistency between abstract & simple summary - are both species endangered? In simple summary, only lynx is cited as endangered.
We have modified the text to indicate that wildcats are considered near threatened, not endangered (lines 32-33): "The wildcat (Felis silvestris) and the Iberian lynx (Lynx pardinus) are important species in Spain, considered as near threatened and endangered, respectively."
Line 35-36: revise as suggested for line 26.
The text has been changed and these words have been removed (lines 34-35): "...a parasite that can cause morbidity and mortality in transplacentally infected or immunocompromised mammals."
Line 40: Remove "The" at beginning of sentence.
Done as requested (line 39): "Indirect Fluroescent Antibody Test (IFAT) ..."
Line 43-44: see comment in General feedback regarding discussion of serology vs molecular testing. Statement should be removed.
As commented above, we think that results from serology and PCR can be compared. Anyway, this paragraph has been removed as suggested.
Following the statement on seroprevalence, should include summary of pcr and fecal testing.
Done as suggested (lines 43-44): "...and parasite DNA was detected in the feces of one wildcat and in tissue samples from 10 wildcats and 11 Iberian lynxes."
Line 45: suggest removing "high presence of toxoplasmosis..." and consider "high risk of infection with T. gondii to animaIs and humans in these areas".
The text has been modified as suggested (lines 44-45): "the epidemiological importance and high risk of T. gondii infection in animals and humans in the studied areas."
Line 45: consider replacing "their" with "feline"
Done as suggested (line 45): " Considering feline susceptibility to infection, ..."
Line 47: consider changing to "feline species"
The text has been modified (line 46): "... to assess the health status of wild felines."
Introduction
Line 53: tropics (not capital).
Corrected (line 52): "... from the Arctic to the tropics, ..."
Lines 54-55: consider revising to “…. a disease of significant zoonotic importance”.
Done as requested (lines 53-54): "... a disease of significant zoonotic importance in both human and animal contexts."
Line 56: consider revising to “…. or can cause severe disease with anorexia …”
The paragraph has been rewritten; this change has been done (lines 55+): "... in immunocompromised or otherwise more susceptible individuals can cause severe disease with anorexia, weight loss, ..."
Line 58: remove “s” from abortions.
Corrected (line 58): "... neurological and systemic disorders, abortion, and even death."
Line 63: consider revising to “… omnivores, infection can occur through ingestion of oocysts or tissue cysts within infected prey.”
The paragraph has been rewritten and the text has slightly changed (lines 64-65): "... herbivores become infected after ingestion mainly of oocysts, whereas infection in carnivores and omnivores can also occur through ingestion of tissue cysts in infected preys."
Line 64: I think it is important to state that felids are the definitive host for this parasite. Therefore, suggest that the sentence begin “As definitive hosts, felines…”
This has corrected as suggested (line 70): "As definitive hosts, felines (both domestic and wild) play a crucial role ..."
Line 65: suggest adding “in all species” after “toxoplasmosis” – to emphasize the role that felines play in disease across all susceptible species.
Done as suggested (line 71): "... in the epidemiology of toxoplasmosis in all species and ..."
Line 67: suggest removing “conducted thus far”
Removed as suggested (line 74): "The majority of epidemiological studies in felines are based ..."
Line 72: suggest adding “however” before “oocysts have not been detected…”
Added as suggested (line 80): "... however, oocysts have not been detected ..."
Line 73: suggest removing “However”.
Removed as suggested: " Epidemiological information is ..."
Line 74: suggest removing “and lacking updated data” as well as “only”
Line 74: suggest adding “in Spain” after “felines”
Changed as suggested (lines 81-82): "Epidemiological information is more limited for the two native species of wild felines in Spain ..."
Line 78: suggest adding “and” after citation at beginning of this line.
Added as suggested (line 86): "... and currently limited to ..."
Line 81: replace “have been” with “were”
Corrected (line 90): "... 1668 individuals were counted in 2022."
Lines 85-87: Suggest that the authors briefly discusses the potential threats to these species – including Toxoplasma as one of them.
The text has been modified (lines 93-95), takin into account the comments made by other reviewers: "There are many potential threats to these species, including habitat destruction and fragmentation, decline of prey, hybridization, hunting, roadkill and infectious and parasitic diseases [30-34]."
Final paragraph: The study objectives should be clearly stated (as done in lines 88-91) and should be the final statement of this paragraph. The manner in which the paragraph is currently written, the additional information strays and distracts the reader. Suggest re-write of this paragraph and consider moving some of the details to a different paragraph in the introduction.
The paragraph has been edited and the final statement has been deleted. Please note that the template available in the journal web site (www.mdpi.com/files/word-templates/animals-template.dot) included some comments about the contents of each section, and in the case of introduction, it says “Finally, briefly mention the main aim of the work and highlight the principal conclusions”. We have noticed now that this part has been removed from the instructions available in the journal web site (https://www.mdpi.com/journal/animals/instructions).
Material & Methods
Line 100: Suggest adding “and animals” to 2.1 subtitle
Done as suggested (line 106): "2.1. Study area and animals"
Line 110: suggest replacing “check-ups” with “evaluations”
Corrected as suggested (line 116): "... captured animals undergoing health evaluations ..."
Line 133: blood samples were obtained directly from the heart in deceased animals – was there a problem with coagulation? If so, do you think that the sample obtained was affected (decreased antibodies due to coagulation?) If yes, perhaps something to be discussed.
Dead animals were frozen as soon as possible by the personnel of the wild recovery centers after death until autopsy was performed, so there was no problem in obtaining blood from the heart. However, as thoracic fluid also was collected, this statement has been added (line 141-143): "Blood samples were collected from live animals through venipuncture, and from deceased animals from the heart; in all cases, thoracic fluid was also collected in case it would be needed." Please note that fluids from carcasses are valid for detection of antibodies (at least, against T. gondii, see Jakubek et al. 012, Acta Vet. Scand. 54: 13).
Line 139: For sake of clarity, may consider indicating that the IFAT was used to assess IgG. Consider making a statement at the beginning of the paragraph to indicate that both IgG and IgM were used to assess seroprevalence. This seems to be unique to other studies that only used one method of assessing serologic status.
The paragraph has been rewritten (lines 151-153) to indicate that the IFAT method was used to assess both IgG and IgM. The kit includes an anti-feline IgG conjugate, but it can be used with an anti-feline IgM conjugate. We have indicated at the end (line 154) that an IgG or IgM titre of ≥1/80 was considered positive: "To detect T. gondii-specific antibodies, the indirect immunofluorescence antibody test (IFAT) was conducted according to method described by [40] using a commercial IFAT kit for anti-T. gondii IgG detection (MegaFLUO® Toxoplasma g., Megacor) following the manufacturer's instructions. The same kit was used in combination with anti-feline IgM conjugate (Fuller Laboratories) to determine IgM levels. The sera were screened at dilutions of 1/40, 1/80, 1/160, 1/320 and 1/640. An IgG or IgM titre of ≥1/80 was considered positive."
Line 146: change “genetic” to “molecular”
Line 146. Replace “In” with “Using”
Changed as requested (lines 157 and 158): "... molecular analysis were also performed. Using the Telemann method, ..."
Line 149: consider replacing “A” with “One”; replace “were” with “was”
Done as suggested (line 160): "One drop of the final sediment was observed ..."
Line 150: change “genetic” to “molecular”
Done as resquested (line 161): "For molecular analysis, ..."
Line 163: remove “s” from DNA and replace “were” with “was”
Done as requested (line 176): "The extracted DNA was then used ..."
Line 169: replace “young” with “juvenile” to keep consistent with line 124.
Done as requested (line 182): "... age groups (juvenile, adult, senior), ..."
Line 173: change “genetic” to “molecular”
Done as requested (line 186): "... and molecular analysis results ..."
Results
Line 176: should consider starting first sentence with “Of the 89 felines sampled, the majority were adults … and there were no juveniles of either species”. Also consider including the demographic information related to the 9 live lynx sampled (age, sex).
In fact, there are no wildcat juveniles, but there are 7 lynxes in this category (see table 3, previous table 5; this table has been modified in accordance with another comment, please see further below). We have not included demographic information on the living separated from the dead because we do not have enough data to make a comparison between them (only 9 alive vs. 80 dead). In addition, in this study we are detailing the infection by T. gondii and in this sense we do not see the importance of separating them.
Lines 179-180, 182, 183, 184: replace “have been” with “were”.
Line 179-185: Since this study is focusing on T. gondii, suggest reporting the oocyst results at the beginning of this paragraph and then summarize other endoparasite results.
This paragraph and table 1 have been deleted, following comments made by other reviewers.
Table 2: replace “title” with “titer”; another way to state titer is 1:80, 1:160, etc
Following other reviewers’ comments, we have change “titer” (American English) by “titre” (British English). We have replaced “title” with “titre” (line 199 and table 1 –former table 2-). In relation to the way to state the titre, the dilutions we have made are “1 in a total volume of 80” (1/80) not “1 and 80 of diluent” (1:80; total volume: 81).
Line 196: change “genetic” to “molecular”
Changed as requested (line 203).
Line 196: consider stating that the sample was positive on all 3 fecal tests
We interpret the reviewer refers to microscopy, qPCR and nPCR. We have modified the text to mention that sample was also positive by microscopy) (lines 203-204): "... parasite DNA was detected in the feces of one wildcat (the individual seropositive for IgM and also IgG and positive by microscopy)... " and state this in a footnote in table 2 (former table 3): "3 One animal tested positive in IFAT (both antibodies), microscopy and molecular analysis (feces and tongue samples)".
Line 201: remove “only”, add “s” to “wildcat”, and add “sample” after “one”
The text has been corrected as suggested (line 208): "... 4 samples from wildcats and one sample ..."
Line 202: suggest adding “PCR” before “methods”
Done as suggested (line 209): "... with both PCR methods."
Table 3. the overall layout is a bit confusing. The footnote is useful for serology results. Consider adding a footnote for coprology results (to indicate the sample positive on all 3 tests). I don’t feel that the combined PCR results for feces is useful and suggest removing from data table. Re: tissue analysis – Suggest removing “Combined” and simply state nPCR + qPCR. Consider adding one final column with heading total PCR positive – which will give the total number of positives by species
We have made some modifications in this table (now table 2). The combined PCR results for feces have been deleted, the results of tissue PCR are given by method, and a final column including all individuals found positive by PCR (any method, any type of sample) has been included.
Line 211: change “genetic” to “molecular”
Following other reviewers suggestions, this table has been removed.
Line 214: consider changing “epidemiological” to “demographic”
Changed as suggested (line 229): "Regarding the analyzed demographic variables associated with ..."
Table 5. The table provides a nice summary of the data. Some suggestions for revision. Replace column headings “n+/n” with simply “n”. In those same columns, state number positive only. If you would like to include the # positive out of the total number of animals (ie. 13/14), consider including this in the % positive column. In the heading “Combined data from both hosts” – consider replacing “host” with “species”.
Changes to the table have been made as suggested. This table is now table 3.
Discussion
Lines 223-225: Consider adding statement made in Lines 235-236 into this first sentence of the paragraph. Also, the first sentence says this is the first study in a large number of wild felines. However, in Table 6, there is a study cited with a larger sample size (n=129). May need to explain or rephrase.
We have changed the text to indicate that this is the first study in a large number of wildcats in Spain (lines 239-241) and of lynxes in Castilla-La Mancha (lines 241-242): "This is the first study to report the prevalence of T. gondii in wildcats in Castilla y León, analyzing a larger number of animals compared to previous studies conducted in other regions of Spain [21,34]. Also, we have now analyzed a greater number of lynxes in Castilla-La Mancha than in previous studies (Table 4)."
Line 232: suggest adding “estimated” before population
Changed as suggested (line 249): "... and the estimated population size has changed, ..."
Line 234-235: may need to further explain the “completely different territorial distribution” OR consider removing “completely” to make it a less strong statement.
To avoid a probably long explanation about the importance of the changes in the distribution of lynxes in the last 10 years, as this is outside the scope of the study, we have decided to remove “completely” (line 252): "... and the estimated population size has changed ..."
Table 6: suggest replacing “this study” with “present study”
Changed as suggested (table 6 is now table 4).
Line 242-243: replace “… fals within the range reported in…” with “is similar to…”
Changed as requested (line 258): "... is similar to other studies on wild felines, including ..."
Line 248, 249: change “genetic” to “molecular”
Changed as requested (lines 262): "... should be confirmed through molecular analysis."
Line 248-253: suggest rewriting to make more succinct. Suggest “All diagnostic results from this (or one) individual are suggestive of active infection with oocyst shedding. This is the first report…”
Changed as suggested (lines 264-265): "All diagnostic results from this individual are suggestive of active infection with oocyst shedding. This is the first report ..."
Line 258: suggest replacing “sera” with “samples”
Changed as suggested (line 282): "... IgG antibodies against T. gondii were detected in 53.9% of the samples."
Line 270: suggest replacing “parasitosis” with “parasite”
Changed as suggested (line 294): "The prevalence of this parasite ..."
Line 280: suggest replacing “feed” with “prey”
Changed as suggested (line 308): "... they may prey on other species ..."
Line 281: replace “hosts” with “species”. Also replace “However” with “in contrast”
Changed as suggested (lines 308-309): ... they may prey on other species (e.g., hares, Lepus sp.). In contrast, the diet of wildcats is ..."
Line 292: double check citation format
We have changed the text to follow the format (line 320): "... and local conditions [13]."
Line 295: suggest to remove “However”
Deleted as suggested (line 323): "... also depends on the susceptibility of each species [61,62]. We have not found ..."
Line 269-295: oocysts can survive well in the environment as evidenced by the global distribution of this parasites and varied climates. However, ideal conditions are considered to be hot, humid climates and lower altitudes (ref: https://www.cdc.gov/parasites/toxoplasmosis/epi.html). This is contradictory to the findings in this study where there was higher seroprevalence in the northern region (described as subhumid, more temperate summer). Therefore, perhaps climate is not a significant factor – as one might expect higher seroprevalence in the animals from the more southern region. When data is combined, it is skewed based on the fact that there was higher overall seroprevalence in the wildcats (almost all from northern region). More likely there is a greater impact of diet – whereby wildcats may be more at risk due to increased number of prey consumed (as you discussed).
We have not entered in a detailed description of the climatic conditions of different Spanish regions, and maybe the descriptions given would lead to missinterpretation as they were made in a comparative way (we stated this in the first manuscript; now it is in lines 295-297 of the revised one). In the northern region of Castilla y León, the climate in summer is hot (up to 40-42ºC) and cold in winter (easily below -10ºC in some areas); rain is usually moderate, with higher values in spring and autumn. In Castilla-La Mancha, summer temps could reach 40-45ºC, while in winter they can drop below -5ºC; precipitation is scarce (this region is part of the so-called “dry Spain”). The conditions in the northern region are better for T. gondii oocyst survival than that in the southern region.
Line 303-306: since there were no juveniles in the study, must be careful making statements regarding age. While the statement was well written, suggest including a statement which reminds the reader that there were no juveniles of either species and only 4 seniors in the lynx population.
As mentioned in a previous response, in fact there are 7 juveniles, all of them lynxes and all of them negative. We have modified the text to clearly state that all wildcats were adults and that the observed increase in prevalence is in lynxes (lines 334-338): "Only adult wildcats have been analysed in this study, but lynxes from all three age groups have been analyzed, although there is a predominance of adult individuals (58) compared to juveniles (7) and seniors (4). In lynxes, we have observed a significant difference among age groups, with an increase in individuals over 1 year old (adults and seniors)."
Line 312: the presence of IgG suggests previous infection/exposure however uncertain if one can assume that all individuals with IgG have tissue cysts. May suggest changing “and” to “with or without tissue cysts”
Please see the response to the first comment. We think it is perfectly valid to consider that all seropositive individuals have tissue cysts, and then it is not necessary to change the text.
Line 312-313: reference is made to feedback in General feedback regarding serology vs PCR. In most cases when comparing these two diagnostic modalities, that the number of PCR positives would be lower than seropositives – as not all individuals with IgG will be actively infected.
Please see the response to the first comment.
Line 316-324: A nice discussion regarding which tissues to sample. It may warrant explanation why higher-yield tissues such as brain and heart were not used in this study and justification why diaphragm was used (as there is no discussion that this was used in other studies)
We were not able to obtain samples from all desired tissues from all road-killed animals; in some cases, the remains were badly injured. Among the organs/tissues that were not available from all animals was the brain. We now have included an statement (lines 367-372) to explain why we have chosen tongue, diaphragm and spleen: "In this study, access to samples of certain tissues from the roadkill animals was not possible, due to the condition in which some of the remains were found. Among the tissues/organs that could not be analyzed in all animals was the brain. The selection of the tongue and spleen was based on the literature, and the choice of the diaphragm was based on the well-oxygenated nature of this muscle."
Line 324: The statement conveys that you assume that the seropositive animals are have tissue cysts. IgG+ individuals may not all have cysts – in this case PCR results are not false negative. Suggest rephrasing to be a more general statement such as “False negatives can occur when …”
Please see the response to the first comment. In any case, we find the suggestion to be valid and it has been incorporated into the text (line 379): "... then, false negatives can occur due to a non-homogeneous distribution ..."
Conclusion
Line 332-335: The first statement on the concluding paragraph should be a summary of the results. The first sentence should be included in the Conclusion but not as the first sentence.
The paragraph has been re-written and the first sentence has changed (lines 387-395): "This is the first study in which T. gondii oocysts were found in fecal samples from wildcat, Felis silvestris, in Spain. Oocysts were not detected in the feces of Iberian lynxes, Lynx pardinus; this is an expected result even in cats, so it is not possible to confirm or deny at this time whether lynxes are definitive hosts of T. gondii. The T. gondii seroprevalence found in wildcats in this study is 85%, and in Iberian lynx it is 45%; in this species, an increase in seroprevalence with the age of the individuals has been observed. The higher seroprevalence in wildcats than in Iberian lynxes may be due to the different geographic locations of the two feline populations, as well as differences in their diet. The detection of the parasite in feces or in tissue samples was low compared to the high seroprevalence detected in the same individuals."
Line 334: add “sero” to “prevalence”
The prefix “sero” has been added in different parts in the text of the conclusions.
Line 340: suggest removing “New”
Deleted as suggested (line 400); "Studies comparing ..."
Line 342: suggest adding justification at end of final sentence, for example “…. may be beneficial for future epidemiological studies.”
Added as suggested (lines 402-403): "... as it may be beneficial for future epidemiological studies."

Reviewer 2 Report
The conclusions show the high value of the results obtained in these wild animals in an environment close to populations. The prevalence detected is surprising if we take into account other studies, but the recovery of the Iberian lynx in Spain makes it necessary to have studies of the relevance of this zoonosis in host species that contribute to the dissemination of the parasite, bringing it closer to other hosts and contributing to the increase of toxoplasmosis. I agree with the affirmation of the need to continue with studies of this nature in order to optimise the most appropriate diagnostic technique and to be able to have an epidemiological map of the disease in Spain.
Author Response
Comments and Suggestions for Authors
The conclusions show the high value of the results obtained in these wild animals in an environment close to populations. The prevalence detected is surprising if we take into account other studies, but the recovery of the Iberian lynx in Spain makes it necessary to have studies of the relevance of this zoonosis in host species that contribute to the dissemination of the parasite, bringing it closer to other hosts and contributing to the increase of toxoplasmosis. I agree with the affirmation of the need to continue with studies of this nature in order to optimise the most appropriate diagnostic technique and to be able to have an epidemiological map of the disease in Spain.
Thanks very much for your comments and the time you have dedicated to reading the paper.

Reviewer 3 Report
This is an interesting epidemiological study on Toxoplasma gondii infection in two wild felids in Spain. However, a number of wrong, incomplete, or missing statements on T. gondii life cycle and infection, on the role this infection may have in these two felid species, and the different roles each of these species may play, are present in the manuscript. Moreover, the aims of the study have been not clearly defined.
Comments and suggested changes are reported below:
Introduction:
Lines 54-55: toxoplasmosis is mostly an infection and not a disease as the majority of T. gondii infections in immunocompetent adult hosts are asimptomatic or sublinical.
Lines 61-64: this is a wrong statement since transplacental transmission is very frequent in many mammal species, such as sheep and goats. Moreover, the milk in neonate mammals is a further important source of infection (and, probably, also the sperm in adult animals).
Line 70: please, explain in detail the acute and cronic phase of T. gondii infection.
Line 87: none of the cited references deal with the role of T. gondii infection in these wild felines. Please explain in detail which role T. gondii may have as a potential threat to these two wild feline species and add appropriate references.
Lines 88-91: epidemiological importance for what? Please explain in detail. Moreover, it should be considered and stated in detail in the manuscript (with references) that, at now, while the wild cat is a recognized definitive host of T. gondii, the Iberian lynx is not. Among felids of the genus Lynx, in fact, only the bobcat (Lynx rufus) has been recognized as a definitive host. Therefore, the aims of the study should be differentiated for these two wild felid species.
Lines 91-98: These are results and should be deleted from the introduction
Materials and methods
Lines 145-150: This is not a standard method for parasitological analysis of fecal samples. Moreover, references used for parasite identification should be added.
Results
Lines 179-184 and Table 1: Capillaria spp. is not correct, since in feline faecal samples only capillariid belonging to the genera Eucoleus and Aonchotheca can be detected. Moreover, Trichuris vulpis infect only canids. More, Ancylostoma spp. should be replaced with Ancylostoma/Uncinaria. Finally, A brief discussion of these results is lacking in the manuscript.
Discussion
Lines 223-224: please replace "in a large number of wild felines" with "in a large number of the two wild feline species considered in the study"
Lines 275-285: please, consider in this section also other infection routes.
Lines 285-288: please add necropsy also in materials and methods and necropsy findings in Results.
Line 306: please, delete "at some point"
Lines 322-324: include that PCR positivity depends also on the number of cysts in a single animal and in a single organ/tissue and that only 2 gr of each organ/tissue were analysed used for PCR analysis.
Finally, a discussion on the significance of the lack of T. gondii oocysts in fecal samples of the Iberian Lynx should be added.
Conclusions
Lines 332-335: this is a wrong and misleading statement as, differently from the wild cat, the Iberian lynx is not a recognized definitive host of T. gondii.
Author Response
Comments and Suggestions for Authors
This is an interesting epidemiological study on Toxoplasma gondii infection in two wild felids in Spain. However, a number of wrong, incomplete, or missing statements on T. gondii life cycle and infection, on the role this infection may have in these two felid species, and the different roles each of these species may play, are present in the manuscript. Moreover, the aims of the study have been not clearly defined.
Firstly, we would like to express our gratitude to the reviewer for the time he/she has dedicated to reviewing this work. In the following lines, we address the comments, accepting some and countering others.
Comments and suggested changes are reported below:
Introduction:
Lines 54-55: toxoplasmosis is mostly an infection and not a disease as the majority of T. gondii infections in immunocompetent adult hosts are asymptomatic or subclinical.
The sentence has been changed (lines 54-58): "Toxoplasma gondii infections in immunocompetent adult hosts are asymptomatic or sublinical, however, in immunocompromised or otherwise more susceptible individuals can cause severe disease with anorexia, weight loss, lethargy, dyspnea, ocular manifestations (chorioretinitis, blindness), neurological and systemic disorders, abortion, and even death. [2]."
Lines 61-64: this is a wrong statement since transplacental transmission is very frequent in many mammal species, such as sheep and goats. Moreover, the milk in neonate mammals is a further important source of infection (and, probably, also the sperm in adult animals).
Thanks very much for taking our attention to this point. In fact, we realize there are several transmission routes but it was not our intention to explain all possibilities (and there are many) in the introduction. We mentioned only the oral route because to the best of our knowledge, it is the only one allowing parasite transmission between different host species, and we were speaking (in introduction) about the transmission between host species depending on the general diet of hosts (herbivores, carnivores/omnivores). The intra-species routes (trans mammary, transplacentary, sperm) can occur in all these types of animals. The sentence has been modified to include possible intra-species transmission (lines 61-67): "The life cycle of T. gondii is complex as a consequence of the diverse routes of transmission between individuals of the same and of different species [5]. Inter-species transmission occurs by the oral route, in which herbivores become infected after ingestion mainly of oocysts, whereas infection in carnivores and omnivores can also occur through ingestion of tissue cysts in infected preys. Intra-species transmission is also possible during fetal development, lactation, or through sperm [6]."
Line 70: please, explain in detail the acute and chronic phase of T. gondii infection.
The acute and chronic phases refer to the developmental stage of the parasite: in the acute phase, the parasite causes intestinal infection with sexual reproduction and oocyst release in feces. In the chronic phase, only tissue cysts with bradyzoites may be present in the infected cats. We have modified the text to indicate these two phases (lines 77-78): "... whether they are in the acute phase (intestinal, with oocyst-release in feces) or in the chronic phase (with tissue cysts)."
Line 87: none of the cited references deal with the role of T. gondii infection in these wild felines. Please explain in detail which role T. gondii may have as a potential threat to these two wild feline species and add appropriate references.
The text has been modified (lines 93-100) and new references have been mentioned to show that despite T. gondii has not been described to date as causing toxoplasmosis in Iberian lynxes or wildcats, the parasite can produce disease in other wild felines as Pallas' cats (Otocolobus manul), sand cats (Felis margarita) and bobcats (Lynx rufus). We think there is no reason to exclude lynxes and wildcats as potentially susceptible hosts, thus justifying the importance of epidemiological studies on these species. The new text is: "There are many potential threats to these species, including habitat destruction and fragmentation, decline of prey, hybridization, hunting, roadkill and infectious and parasitic diseases [30-34]. Within parasitic infections, those caused by Toxoplasma gondii in wild felines seem to be rarely associated with disease, and no cases of clinical toxoplasmosis have been described in Iberian lynx or wildcats. However, clinical cases have been reported in Pallas's cats (Otocolobus [Felis] manul), and neonatal toxoplasmosis has been documented in sand cats (Felis margarita) and bobcats (Lynx rufus) [35-37]."
Lines 88-91: epidemiological importance for what? Please explain in detail. Moreover, it should be considered and stated in detail in the manuscript (with references) that, at now, while the wild cat is a recognized definitive host of T. gondii, the Iberian lynx is not. Among felids of the genus Lynx, in fact, only the bobcat (Lynx rufus) has been recognized as a definitive host. Therefore, the aims of the study should be differentiated for these two wild felid species.
It has long been recognized (see references 5 and 9) that oocysts shed by domestic cats can be responsible for epidemic outbreaks. However, there is no data on the potential significance of wild felids, which will depend, among other factors, on the prevalence of infection in these animals. We think the new redaction in lines 71-72 indicates the importance of domestic felids in the epidemiology of the parasite: "... As definitive hosts, felines (both domestic and wild) play a crucial role in the epidemiology of toxoplasmosis in all species and can be responsible (at least, domestic cats) of outbreaks [5,9]." This justify (lines 101-104) the present study to assess if wild felids in Spain may have epidemiological importance in T. gondii transmission: "The objective of the present study has been to evaluate the presence and prevalence of T. gondii in wild populations of Iberian lynx and wildcat in Spain, in order to assess the epidemiological importance of these hosts and evaluate the potential impact of the parasite on the conservation of these protected species."
Regarding the comment excluding Iberian lynx as definitive hosts of T. gondii, we believe that it is correct from a strictly purist standpoint, but it does not correspond to reality. There are around 20 confirmed felid species that shed oocysts. It is true that T. gondii oocysts have not been found in the feces of Iberian lynx in this or other studies, which is a challenging finding due to the short period during which oocysts are shed in the feces of definitive hosts. However, this does not imply that lynxes are not definitive hosts. In fact, it is not common to find oocysts in the feces of domestic cats. The diversity of genera that include definitive hosts, Lynx among them, suggests that Iberian lynx should be considered as a potential and most likely real definitive host of T. gondii. We have added some text in the discussion to specify that oocysts have been found in wildcats but not in Iberian lynx (line 265-268), and that the negative results for oocyst detection in feces by microscopy and PCR in lynxes do not allow to propose this species as definitive host of T. gondii, although in our opinion it should be considered a potential and probably valid definitive host of this parasite (lines 268-273). The new text (lines 265-273) is: "This is the first report in Spain of T. gondii oocysts in a wild feline (wildcat) in its natural environment. In the case of Iberian lynx, the negative results for the presence of oocysts in feces or from PCR analysis of fecal samples prevent confirming this species as a definitive host of T. gondii. This negative result is not surprising; due to the short patent period, it is difficult to detect oocysts even in cat feces [52]. However, the diversity of felid species accepted as definitive hosts, including congeneric species (L. lynx, L. rufus) [7,8] on that list, strongly suggests that the Iberian lynx may also be a definitive host. Further studies are needed to undoubtedly confirm this possibility."
Taking this possibility into account, the aim of this study is to assess the importance that lynxes and wildcats may have in the epidemiology of T. gondii by determining the presence and prevalence* of the parasite in both host species (by detecting oocysts in feces, tissue cysts by PCR, and antibodies in serum). We have modified the text (line 101: "The objective of the present study has been to evaluate the presence and prevalence of T. gondii ...") to clarify the objective of the study, which this way is the same for both wild felid species.
* The reviewer may question that presence and prevalence can be determined by detecting antibodies in serum, as antibodies indicate previous contact but not necessarily present infection. This problem has been pointed out by other reviewers and we have explained why we think that seropositive animals are true infected animals. The infectivity of bradyzoites in cats is very high, with as few as 50 bradyzoites (range 2-181) being sufficient to produce tissue cysts and intestinal infection (Cornelissen et al., 2014, PLOS One 9: e104740) and 100 oocysts to produce tissue cysts (Dubey 1997, Parasitology 115: 15-20). Considering that a tissue cyst harbors between 2-1000 bradyzoites (Dubey et al., 1998, Clin. Microbiol. Rev. 11:267-299), it can be assumed that a cat (and by extension, other felids) feeding on infected intermediate hosts (the normal route of transmission) or ingesting enough number of oocysts (100, maybe less) will develop tissue cysts. Tissue cysts will survive for several years, and although it is not definitively confirmed, it is assumed they would survive for the entire lifespan of the host, Then, we consider seropositive animals as truly infected and harboring tissue cysts.
Lines 91-98: These are results and should be deleted from the introduction
These lines have been removed as requested. Please note that the template available in the journal web site (www.mdpi.com/files/word-templates/animals-template.dot) included some comments about the contents of each section, and in the case of introduction, it says, “Finally, briefly mention the main aim of the work and highlight the principal conclusions”. This part has been removed from the instructions available in the journal web site (https://www.mdpi.com/journal/animals/instructions).
Materials and methods
Lines 145-150: This is not a standard method for parasitological analysis of fecal samples. Moreover, references used for parasite identification should be added.
We disagree with the reviewer about the validity of the Telemann method for the routinary analysis of fecal samples. It is true that floatation methods are not recommended when some parasitic structures are suspected to occur (i.e., operculated eggs, thin-shelled eggs), but the method is perfectly valid for oocyst identification. In fact, the modified Telemann method is especially useful for samples with high fat concentrations (like those from carnivores). We have added a new reference to support our opinion (line 156): "The fecal samples were processed using the modified Telemann method [41,42] ..."
Results
Lines 179-184 and Table 1: Capillaria spp. is not correct, since in feline faecal samples only capillariid belonging to the genera Eucoleus and Aonchotheca can be detected. Moreover, Trichuris vulpis infect only canids. More, Ancylostoma spp. should be replaced with Ancylostoma/Uncinaria. Finally, A brief discussion of these results is lacking in the manuscript.
At the request of another reviewer's comment, the results of the other parasites detected in the stool analysis have been removed.
Discussion
Lines 223-224: please replace "in a large number of wild felines" with "in a large number of the two wild feline species considered in the study"
This sentence has been deleted in the new paragraph, which has been rewritten (lines 239-252): "This is the first study to report the prevalence of T. gondii in wildcats in Castilla y León, analyzing a larger number of animals compared to previous studies conducted in other regions of Spain [21,34]. Also, we have now analyzed a greater number of lynxes in Castilla-La Mancha than in previous studies (Table 4). Furthermore, the availability of animals for epidemiological studies in these species is typically low and limited to specific territories. Most epidemiological studies on T. gondii in Iberian lynx have focused on populations in Andalusia [22-24], with only one study conducted in Castilla-La Mancha [21]. Similarly, studies on the wildcat have been limited to a small number of animals (6 to 9 individuals) from the Cantabrian Mountains [21] and Castilla-La Mancha [34]. Our study provides updated data on T. gondii infection in the Iberian lynx, considering that more than 10 years have passed since the last published studies, and the estimated population size has changed, increasing from 213-241 individuals (2008-2009) to over 1500 individuals currently [29]. This increase in population size results in greater interaction among individuals and a different territorial distribution."
Lines 275-285: please, consider in this section also other infection routes.
The text has been modified and the possibility of infection by ingestion of oocysts has been added (lines 301-305): "In the case of T. gondii, the definitive host can become infected through the ingestion of oocysts (whose viability depends on environmental factors). However, although possible, the ingestion of such a high number of oocysts would be required, making this route of infection highly inefficient [57]; it is considered that the parasite is primarily transmitted to the definitive host through predation [6,57]."
Lines 285-288: please add necropsy also in materials and methods and necropsy findings in Results.
No lesions were detected in the necropsies. Following the same criterion applied to remove the table about parasite findings in the coprological analysis, necropsies and their (negative) findings have not been included. Besides, necropsies were not made by the authors but by veterinarians authorized by the Regional Environment Departements of the Autonomous Communities in which the study were performed. In some cases, one of the authors (Pablo Matas) was present as an observer in some of the autopsies done by the veterinarians, this allowing the comment made in lines 314-315 as “personal observation”: "... while in the stomachs of lynxes, which are strictly dependent on European rabbits, remains of 1-2 rabbits were found (P. Matas Méndez, personal observation)."
Line 306: please, delete "at some point"
Done as requested (line 339).
Lines 322-324: include that PCR positivity depends also on the number of cysts in a single animal and in a single organ/tissue and that only 2 gr of each organ/tissue were analysed used for PCR analysis.
Done as requested (lines 376-379): "... as the definitive host also has a higher probability of having been infected with increasing age ..."
Finally, a discussion on the significance of the lack of T. gondii oocysts in fecal samples of the Iberian Lynx should be added.
Done as requested (lines 388-390): "Oocysts were not detected in the feces of Iberian lynxes, Lynx pardinus; this is an expected result even in cats, so it is not possible to confirm or deny at this time whether lynxes are definitive hosts of T. gondii..
Conclusions
Lines 332-335: this is a wrong and misleading statement as, differently from the wild cat, the Iberian lynx is not a recognized definitive host of T. gondii.
As mentioned in a previous response, we consider Iberian lynx as a potential, probably true definitive host of T. gondii. Anyway, the text has been modified (lines 388-390): "Oocysts were not detected in the feces of Iberian lynxes, Lynx pardinus; this is an expected result even in cats, so it is not possible to confirm or deny at this time whether lynxes are definitive hosts of T. gondii."

Reviewer 4 Report
The manunscript entitled "Prevalence of Toxoplasma gondii in endargered wild felines (Felis silvestris and Lynx pardinus) in Spain provides updated information prevalence and seroprevalence of T. gondii infection in wild felines that deserves to be published.
However several main issues need to be corrected before be accepted.
- The topic of the manuscript is the prevalence of Toxoplasma gondii infection in wild felines. It does not make sense to include the results of nematodes and other endoparasites due to they do not contribute anything to the main topic of the manuscript. It seems that these results are "included by force" due to the authors only show the results in a table but they do not comment anything about nematodes in the introduction neither in the discussion. I recommend to remove it from the manuscript.
- For an accurate diagnosis, more than one technique is required, above all in wildlife samples. The authors have used two different PCRs, however they have only used one serological technique. I recommend using an other serological technique such as ELISA or MAT.
- The genotype of T. gondii provides valuable information for epidemiological studies due to differences regading its virulence. It would be a good option to improve this manuscript include the genotyping of the positive samples.
Minor changes:
Introduction.
- Lines 25, 85, and 90. The assumption that T. gondii infection in cats can weaken the immune system and affect their survival is incorrect. Felines are the definitive host of T. gondii and chronic infected cats are asymptomatic. In fact, the authors reference some studies to state that T. gondii is able to weaken their immune system, however, in that studies (e.g. reference 27) the authors only state that the animals had the disease and they do not state nothing else. Please, remove theses assumptions thoroughout the manuscript.
Line 62: the assupmtion that "in herbivores , infection occurs solely through the ingestion of occysts is incorrect". T. gondii infectio has been proved through oral ingestion of tissue cysts (see: Verhelst et al., 2014). Please, rephrase.
- Authors need to explain to the readers the geographical distribution and census of both species in the Iberian Peninsula before sampling. I suggest to include in the map of figure 1 this data togehter with the data of sampling collection.
Material and methods
Line 101. Please provide information about census and geographical distribution
Line 132. Sampling the blood from the heart in dead animals is a difficult task as most of the times is coagulated. Thus, the second option is sampling the thoracic fluid. Can the authors specify how the did it? How it was the blood from at the moment of sampling? How many hours/days was dead before sampling?
Line 137: the CNS is one of the main target samples to carried out the diagnosis of toxoplasmosis. Please, explain why the authors have decided to sampling tongue, diaphragm and spleen instead of others. Are convenience samples?
Line 156: Please, provide the dilution of DNA to perform the PCRs.
Lines 158-160: Something is missing in the editing regarding the reference 36. Please rephrase.
- Line 166: Please, specify if Se and Sp of serological test were considered to calculate the seroprevalence rate.
Results
-Lines 196-198: I think date of positive samples do no mach between text and table information. In line 197 the authors state that 10 wild cats and 11 Iberian lynxes were positive for tissue samples but in the table 3 I only see 6 and 10, respectively. Please, check it.
Table 4. Authors need to shorten the information. Combination of positives between organs are not necessary. Please, remove it
Line 219: the authors show that Burgos and Toledo were the provinces with the highest seroprevalence rates. These provinces were the most sampled ones. This issue need to be considered in the discussion section.
Discussion
The authors state that this is the first study on the prevalence of T. gondii in the Iberian Peninsula but they do not provide any information about studies carried out in Portugal. Any information about it? If there are not studies in Portugal, please, indicate.
Line 246: Please, remove studies with captive animals in zoos because they are not comparable.
Regarding the discussion, it would be interesing to compare the prevalence or the seroprevalence of wild felines to that one in other hosts such as rodents, rabbits and ruminants that act as intermediate host. Please include this information with the available data carried out in Spain.
Editing of english language is required
Author Response
Comments and Suggestions for Authors
The manunscript entitled "Prevalence of Toxoplasma gondii in endargered wild felines (Felis silvestris and Lynx pardinus) in Spain provides updated information prevalence and seroprevalence of T. gondii infection in wild felines that deserves to be published.
Thanks very much for the time dedicated to the review of this article and for your favorable opinion.
However several main issues need to be corrected before be accepted.
- The topic of the manuscript is the prevalence of Toxoplasma gondii infection in wild felines. It does not make sense to include the results of nematodes and other endoparasites due to they do not contribute anything to the main topic of the manuscript. It seems that these results are "included by force" due to the authors only show the results in a table but they do not comment anything about nematodes in the introduction neither in the discussion. I recommend to remove it from the manuscript.
These results have been removed. They were included to preempt comments from reviewers questioning why coprological analyses were conducted but no mention was made of findings related to other parasites that may have been found.
- For an accurate diagnosis, more than one technique is required, above all in wildlife samples. The authors have used two different PCRs, however they have only used one serological technique. I recommend using an other serological technique such as ELISA or MAT.
We agree the reviewer that the use of two or more techniques would be advisable for an accurate diagnosis. However, except when comparing different methods or commercial kits, in most studies only one method is used (apart of those works listed in table 3, see, i.e., Sioutas et al., Pathogens 2022, 11:1511; or the reviews by Montazeri et al., Parasit. Vectors 2020, 13:82, and Galeh et al. 2020, Front. Vet. Sci. 7:461). Among the different methods used in other studies (ELISA, MAT, IFAT, IHA, immunochromatographic tests), IFAT has the same sensitivity and specificity as ELISA for the detection of T. gondii (Sharma et al., Food Waterborne Parasitol. 2019, 12:e00046).
We have used two different PCR techniques (nested PCR and real-time PCR) that amplify two different gene targets to increase the sensitivity in tissue samples, in which it can be difficult to obtain positive results due to the probable low number of cysts.
- The genotype of T. gondii provides valuable information for epidemiological studies due to differences regading its virulence. It would be a good option to improve this manuscript include the genotyping of the positive samples.
We agree the reviewer about genotyping the T. gondii isolates can provide valuable information. For this purpose, isolation is necessary to obtain a sufficient and pure amount of T. gondii DNA to perform the genotyping. In our samples, however, parasite isolation was not possible because samples were freeze by the veterinarians of the Regional Environmental Departments before they were provided to us (we have now stated this in the text, line 124). We are currently conducting studies attempting to characterize the parasite directly from the samples of positive animals but we do not yet have results.
Minor changes:
Introduction.
- Lines 25, 85, and 90. The assumption that T. gondii infection in cats can weaken the immune system and affect their survival is incorrect. Felines are the definitive host of T. gondii and chronic infected cats are asymptomatic. In fact, the authors reference some studies to state that T. gondii is able to weaken their immune system, however, in that studies (e.g. reference 27) the authors only state that the animals had the disease and they do not state nothing else. Please, remove these assumptions throughout the manuscript.
The paragraph in the introduction has been rewritten (lines 54-58) and the assumption that T. gondii infection in cats can weaken their immune system has been removed: " Toxoplasma gondii infections in immunocompetent adult hosts are asymptomatic or sublinical, however, in immunocompromised or otherwise more susceptible individuals can cause severe disease with anorexia, weight loss, lethargy, dyspnea, ocular manifestations (chorioretinitis, blindness), neurological and systemic disorders, abortion, and even death [2]."
In the Summary and Abstract, the text has been modified to note that the parasite can affect the survival of immunocompromised hosts (lines 26-27): "Both feline species can become infected by Toxoplasma gondii, a parasite that can cause morbidity and mortality in transplacentally infected or of immunocompromised mammals." (lines 33-35): "Both can be infected by Toxoplasma gondii, a parasite that can cause morbidity and mortality in transplacentally infected or immunocompromised mammals."
Line 62: the assupmtion that "in herbivores, infection occurs solely through the ingestion of occysts is incorrect". T. gondii infection has been proved through oral ingestion of tissue cysts (see: Verhelst et al., 2014). Please, rephrase.
Thanks for noting this point. The text has been corrected to indicate that infection occurs in herbivores mainly by ingestion of oocysts (line 64): "... herbivores become infected after ingestion mainly of oocysts, ..."
- Authors need to explain to the readers the geographical distribution and census of both species in the Iberian Peninsula before sampling. I suggest to include in the map of figure 1 this data togehter with the data of sampling collection.
The map in figure 1 has been splitted into two new maps, one for each host. In each map, the geographical distribution of the species, its census, and the locations where samples were collected, are indicated. In order to prevent the maps from being overloaded with information, we have removed the number of samples from each province, which are indicated in Table 3 (previously, Table 5).
Material and methods
Line 101. Please provide information about census and geographical distribution.
Thie geographical distribution is now provided in the maps in figure 1. The most recent census information on Iberian lynxes is indicated in the text (line 89): "... 1668 individuals were counted in 2022 ...". We have not found official data on wildcat populations.
Line 132. Sampling the blood from the heart in dead animals is a difficult task as most of the times is coagulated. Thus, the second option is sampling the thoracic fluid. Can the authors specify how the did it? How it was the blood from at the moment of sampling? How many hours/days was dead before sampling?
The roadkill animals were collected by environmental officers and subsequently frozen in wildlife recovery centres in less than 24 hours from the estimated time of dead, until autopsies were conducted by the authorized veterinarians from Regional Environmental Departments. During the autopsy, the veterinarians opened the heart to take the clots lodged in the right ventricle and put them in specific tubes without additives to obtain the serum by centrifuging at 1500 rpm for 10 minutes. The blood collected was in cavities as a consequence of capillary exudate due to the process of freezing and thawing of the animal. This affects the permeability of the blood vessels, so that a certain amount of blood can flow out.
Line 137: the CNS is one of the main target samples to carried out the diagnosis of toxoplasmosis. Please, explain why the authors have decided to sampling tongue, diaphragm and spleen instead of others. Are convenience samples?
According to Dubey (2021) [1], T. gondii was more prevalent in muscles than in the brain. Brain samples were not accessible in all road killed animals; in part of those where it could be obtained, the brain was partially degenerated. However, muscle samples (from tongue and diaphragm) were accessible in all individuals and were not degenerated.
Line 156: Please, provide the dilution of DNA to perform the PCRs.
In the nested PCR technique, 5 and 15 ul of DNA from each sample were processed. In the qPCR 5 ul of DNA. This is included in the text in the material and methods section. These data is now provided in the text (lines 165-166): "... following the protocol described by [43] by using 5 and 15 µl of DNA from each sample ..."; (line 168): "... by using 5 µl of DNA from each sample."
Lines 158-160: Something is missing in the editing regarding the reference 36. Please rephrase.
We have corrected the phrase: “Following the protocol described by [45], …” (line 172).
- Line 166: Please, specify if Se and Sp of serological test were considered to calculate the seroprevalence rate.
Sensibility and specificity were not considered to calculate the seroprevalence rate, because they cannot be determined with present data. To determine them, it is necessary to know when an individual is in fact positive, being this the reference for comparing the serology results.
Results
-Lines 196-198: I think date of positive samples do no mach between text and table information. In line 197 the authors state that 10 wild cats and 11 Iberian lynxes were positive for tissue samples but in the table 3 I only see 6 and 10, respectively. Please, check it.
The values are correct, anyway we have changed the way they are presented. There have been 10 wildcats and 11 Iberian lynxes that were positive in the DNA assays. In table 3, we indicate that 4 wildcats were positive by nPCR (only), 2 by qPCR (only), and 4 by both nPCR and qPCR (10 in total); the same for lynxes. In the new table 2, we now indicate how many samples were positive by nPCR, how many by qPCR, and how many samples in total were positive by DNA analysis (some samples were positive by both methods).
Table 4. Authors need to shorten the information. Combination of positives between organs are not necessary. Please, remove it
Following reviewer suggestion, the table has been removed. There is only one mention to the data it showed and it has been changed to “data not shown” (line 214).
Line 219: the authors show that Burgos and Toledo were the provinces with the highest seroprevalence rates. These provinces were the most sampled ones. This issue need to be considered in the discussion section.
Thank you very much for bringing this to our attention, as we made an error in the original statement. Instead of saying "seroprevalence," we should have said "number of samples." The sentence has been corrected (line 233): "The provinces with the highest number of samples were ..." We have included new text about the prevalence in relation to geographic distribution in the discussion (lines 340-353): "Regarding the geographical distribution, the significant disparity in the number of wildcats collected across different provinces, with only 1-3 animals sampled in most of them, precludes drawing conclusions regarding this variable in this host species. Furthermore, there is no reliable population census available, as wildcat is not considered an endangered species but rather classified as near-threatened, which diminishes official interest and the publication of studies. In the case of Iberian lynx, the samples primarily originate from two areas, one in the province of Toledo and another in Ciudad Real, corresponding to the distribution of the populations reported in the latest census [29]. In the province of Albacete, only 2 lynxes have been analyzed; however, considering that the estimated population in this province was 6 individuals, it represents a significant percentage of animals. Overall, the observed seroprevalence for lynx is similar across the different studied areas, averaging around 45%. This value is at the lower limit of the range reported in other studies in Spain (table 4). To the best of our knowledge, there are not available published data from Portugal."
Discussion
The authors state that this is the first study on the prevalence of T. gondii in the Iberian Peninsula but they do not provide any information about studies carried out in Portugal. Any information about it? If there are not studies in Portugal, please, indicate.
To the best of our knowledge, there are no published data on T. gondii prevalence in wildcat or Iberian lynx in Portugal. We indicate this in the text (lines 352-353): "To the best of our knowledge, there are not available published data from Portugal."
Line 246: Please, remove studies with captive animals in zoos because they are not comparable.
Done as requested (lines 258-261): "... other studies on wild felines, including red lynx (Lynx rufus) (6.2%; 1/16) and cougars (Puma concolor) (1.9%; 1/52) in North America [8], jaguars (Puma yagouaroundi) (1.22%; 1/82) in a regional park in Brazil [47], and wildcats (F. silvestris) from Greece (1.6-4.3%)[48]."
Regarding the discussion, it would be interesting to compare the prevalence or the seroprevalence of wild felines to that one in other hosts such as rodents, rabbits and ruminants that act as intermediate host. Please include this information with the available data carried out in Spain.
We have included the data requested (lines 323-328): "We have not found published studies on the seroprevalence of T. gondii in rodents in Spain; in other European countries, it varied between 0-59.4% [63]. In other host species from Spain, seroprevalence varied between 5.6-39.6% in ruminats (roe deer, fallow deer, red deer, Sothern chamois, mouflon, wild goat and Barbary sheep) [64], 21.9% in free ranging Iberian pigs [65], and 6.1-53.8% in wild rabbits [66]."

Reviewer 5 Report
It was very interesting to read the article on Toxoplasma in endangered wild felids in Spain. It is rare to see studies carried out on wild felids, and as such this is a valuable addition to the field. I hope that the authors have further plans to use the valuable samples for detection of other pathogens too.
I was however somewhat surprised not to see a mention of how road traffic accidents may have skewed the data. Toxoplasma is a known behavioural alteration pathogen, and affects risk taking in humans, primates and rodents, and probably too in felids. Is it therefore possible that the prevalence seen is higher due to the samples being obtained from traffic accident cats? It would be nice to see a comparison between the caught vs the killed animals to assess the pathogen prevalence and a discussion of how these samples may skew results.
Other minor comments
Line 88-98- this seems more like discussion than an aims section and may be better moved there?
Table 2- is it titre rather than title?
Line 314- could environmental challenge also lead to an increase in seroprevalence without the development of cysts in the animals?
But overall, an interesting, and well written study, so I congratulate the authors
Very minor suggestions are made above
Author Response
Comments and Suggestions for Authors
It was very interesting to read the article on Toxoplasma in endangered wild felids in Spain. It is rare to see studies carried out on wild felids, and as such this is a valuable addition to the field. I hope that the authors have further plans to use the valuable samples for detection of other pathogens too.
Thanks very much for the time the reviewer has dedicated to this work and for his/her favorauble opinion.
I was however somewhat surprised not to see a mention of how road traffic accidents may have skewed the data. Toxoplasma is a known behavioural alteration pathogen, and affects risk taking in humans, primates and rodents, and probably too in felids. Is it therefore possible that the prevalence seen is higher due to the samples being obtained from traffic accident cats? It would be nice to see a comparison between the caught vs the killed animals to assess the pathogen prevalence and a discussion of how these samples may skew results.
This is a very interesting point. It is accepted that the parasite can influence the behaviour of the intermediate host, and the Fatal attraction phenomenon (changing the fear to the odor of the definitive host into an attraction to this odor) may be the most impressive manipulation. Other behavioural changes may lead to an increased risk of being road-killed. However, in our study, 7 out of the 9 live (captured) lynxes were positive (~78%), while 24 out of 60 road killed animals (40%) were positive. We realize that the number of captured lynxes is low (only 9) and no definitive conclusions can be drawn (for this reason, we have not mentioned this in the manuscript); however, these numbers indicate that T. gondii prevalences are not related to road accidents.
Other minor comments
Line 88-98- this seems more like discussion than an aims section and may be better moved there?
The text has been removed as requested. Please note that the template available in the journal web site (www.mdpi.com/files/word-templates/animals-template.dot) included some comments about the contents of each section, and in the case of introduction, it says “Finally, briefly mention the main aim of the work and highlight the principal conclusions”. This part has been removed from the instructions available in the journal web site (https://www.mdpi.com/journal/animals/instructions).
Table 2- is it titre rather than title?
We have revised the text and have changed titer (American English) -> titre (British English). The error in the old table 2 (now, table 1) has been corrected.
Line 314- could environmental challenge also lead to an increase in seroprevalence without the development of cysts in the animals?
This is another very good question. In theory, there could be cases of individuals in which T. gondii has not developed into tissue cysts because innate immunity has eliminated it before, but antibodies have formed due to activation of adaptive immunity. In this case, prevalence (results from molecular analyses) and seroprevalence would be very different concepts, as prevalence would indicate the number of true parasitized individuals in the population, while seroprevalence would indicate the number of individuals who have been infected at some moment in the past but may not be currently parasitized. However, the infectivity of bradyzoites in cats is very high, with as few as 50 bradyzoites (range 2-181) being sufficient to produce tissue cysts and intestinal infection (Cornelissen et al., 2014, PLOS One 9: e104740) and 100 oocysts to produce tissue cysts (Dubey 1997, Parasitology 115: 15-20). Considering that a tissue cyst harbors between 2-1000 bradyzoites (Dubey et al., 1998, Clin. Microbiol. Rev. 11:267-299), it can be assumed that a cat (and by extension, other felids) feeding on infected intermediate hosts (the normal route of transmission) or ingesting enough number of oocysts (100, maybe less) will develop tissue cysts. Then, we consider seropositive animals as truly infected and harboring tissue cysts, and then prevalence (results from molecular analyses) and seroprevalence are equivalent.
But overall, an interesting, and well written study, so I congratulate the authors
Thank you very much.
